# Association of long COVID documentation with clinical outcomes among Veterans with diabetes

**Pandora L. Wander** [1,2]\*, **Elliott Lowy** [1], **Anna Korpak** [1], **Lauren A. Beste** [1,2], **Edward J. Boyko** [1,2]

**1** Veterans Affairs (VA) Puget Sound Health Care System, Seattle, Washington, United States of America,
**2** Department of Medicine, University of Washington, Seattle, Washington, United States of America

\* lwander@uw.edu

## Abstract

### Objective

To examine public health impacts of Long COVID on long-term hyperglycemia and metabolic health.

### Materials & Methods

We conducted a retrospective cohort study using US Veterans Health Administration electronic health records data to examine associations of Long COVID documentation (International Statistical Classification of Diseases, Tenth Revision code U09.9) with clinical outcomes (number of primary care visits, receipt of new classes of glucose-lowering medications, weight change, hemoglobin A1c, initiation of insulin, initiation of dialysis, hospitalization, and mortality) among U.S. Veterans with diabetes (1 October 2021–1 October 2023; n = 1,896,080).

### Results

Veterans were 69.7 years old on average at cohort entry, 6% female, and 1% had U09.9 documentation. Compared to Veterans without U09.9, those with U09.9 had 39% more primary care visits per year after the index date (incidence rate ratio [IRR] 1.36, 95%CI 1.36; 1.37), 21% more glucose-lowering medication classes added per year (IRR 1.21, 95%CI 1.18; 1.24), a 0.62 kg greater weight gain (95%CI 0.52; 0.72), a 0.10-point lower mean HbA1c (95%CI -0.12; -0.08), a 43% greater odds of starting insulin (odds ratio [OR] 1.43, 95%CI 1.32; 1.54), a 34% greater odds of starting dialysis (OR 1.34, 95%CI 1.11; 1.62), a 102% greater odds of VA hospitalization (OR 2.02, 95%CI 1.95; 2.09), and a 13% lower odds of mortality (OR 0.87, 95%CI 0.83; 0.91).

### Conclusions

In Veterans with diabetes, Long COVID documentation was associated with greater medication use, odds of starting dialysis, and odds of hospitalization, but with lower

**Data availability statement:** Data cannot be shared publicly as a condition of approval by the VA Human Research Protection Program at VA Puget Sound. Inquiries may be directed to the VA Puget Sound Research & Development Program at vapugetsoundresearch@va.gov. Data may be available for VA researchers with appropriate approvals.

**Funding:** Funded by a seed grant from the VAPSHCS Office of Research & Development. Salary support was provided to PLW, EL, AK, LAB, and EJB by VAPSHCS and/or the Office of Research & Development. The study sponsor/ funder was not involved in the design of the study; the collection, analysis, and interpretation of data; or writing the report; and did not impose any restrictions regarding the publication of the report.

**Competing interests:** The authors have declared that no competing interests exist.

odds of mortality. Individuals with Long COVID documentation did not have more weight gain or higher HbA1c, suggesting that adverse effects of Long COVID on medication changes and kidney function in persons with diabetes may be due to other factors. Future studies should examine mechanisms by which Long COVID might contribute to progression of kidney disease and more intensive diabetes treatment.

## Introduction

Long COVID (estimated by the presence of the International Statistical Classification of Diseases, Tenth Revision (ICD-10) code U09.9, "post-COVID condition, unspecified") occurs in roughly 5% of U.S. Veterans after a positive test for SARS-CoV-2 [1]. Autonomic dysfunction and post-exertional malaise are predominant features in Long COVID [2] and may reduce the ability to engage in physical activity. Deficits in executive function are also common [3] and may affect the ability to perform tasks such as glucose testing and decisions about how much insulin to take with downstream effects on diabetes management. Effects of such Long COVID sequelae on long-term hyperglycemia and metabolic health in persons with diabetes are unknown. Additionally, given that COVID has previously been linked with new-onset diabetes [4,5], mechanisms that might contribute to development of diabetes might also drive worsening hyperglycemia in individuals with prevalent diabetes,

Because of these potential impacts on physical activity and cognition, we hypothesized that Long COVID might contribute to adverse clinical outcomes in diabetes including more primary care utilization, need for additional glucose-lowering medications, weight gain, increases in hemoglobin A1c, initiation of insulin, initiation of dialysis, hospitalization, and higher mortality. Further, because insulin use necessitates glucose monitoring, and in some cases dose adjustments and/or timing of injections relative to meals and physical activity, the cognitive demands required to manage insulin use may be relatively greater than the demands to manage diet and non-insulin glucose-lowering medications [6]. Thus, we examined whether associations of Long COVID with clinical outcomes were greater among insulin users than non-users. We conducted a retrospective observational cohort study using Veterans Affairs (VA) Veterans Health Administration (VHA) electronic health record (EHR) data to determine whether the presence of a diagnosis code for Long COVID (i.e., U09.9) is associated with adverse outcomes among Veterans with diabetes.

## Materials & methods

### Study setting and population

We conducted this analysis using the Corporate Data Warehouse (CDW), a data repository derived from VHA's electronic medical record, the largest national integrated health care system in the U.S [7]. We identified all enrollees with prevalent diabetes, defined by (1) two or more abnormal lab values from plasma or serum (random glucose ≥200 mg/dL, fasting glucose ≥126 mg/dL, two-hour glucose from an oral glucose tolerance test ≥200 mg/dL) or whole blood (A1c ≥ 6.5%) [8]; or (2)

ICD-10 codes of E08-E13; or (3) receipt of an initial and one refill prescription of a glucose-lowering medication between October 1, 2021 (the date the U09.9 ICD-10 code was established) and October 1, 2023. Participants were also required to be active in VA care, defined by the presence of ≥1 inpatient or outpatient visit in the prior 12 months. Because prior COVID is highly prevalent (>41% of U.S. adults by serology in 2022 [9]) and testing patterns are not consistent, in order to increase external validity of findings, we did not require documentation of a prior positive COVID test. For individuals with Long COVID documentation the index date was defined as the date of the first instance of U09.9 to appear in the medical record. Individuals without Long COVID documentation were matched by month to those with Long COVID documentation by the presence of a clinic visit or inpatient discharge. For individuals without Long COVID documentation, the index date was defined as a randomly selected date during the month in which the clinic visit or discharge occurred.

## Variables

We extracted data on clinical and demographic characteristics from the medical record including documentation of Long COVID, age, sex at birth, race/ethnicity, Hemoglobin A1c, body mass index (BMI), number of outpatient primary care visits in the year prior, urban/rural residence, SARS-CoV-2 vaccination, and VA facility location. We assessed for the presence of chronic comorbidities using the Charlson comorbidity index (which includes history of myocardial infarction, heart failure, peripheral vascular disease, stroke/transient ischemic attack, dementia, chronic obstructive pulmonary disease, connective tissue disease, peptic ulcer disease, liver disease, chronic kidney disease, solid tumor, leukemia, lymphoma, or acquired immune deficiency syndrome) [10], as well as hypertension and dyslipidemia, which were identified using ICD-9 and ICD-10 codes), Documentation of Long COVID was defined by the appearance of ≥1 instance of U09.9, as we have done previously [1]. Veterans specified any number of race/ethnicity responses, which were categorized as selected/not selected for each individual response (e.g., white yes/no). Mean HbA1c was calculated using all values in the year prior to the index date. BMI was calculated using the most recent weight from the prior two years and the most recent height from the prior five years. Facility location was classified using Veterans Integrated Service Networks (VISN), which comprise 18 regional systems of care within VHA.

For the clinical outcomes, we extracted data on the number of primary care visits after the index date, receipt of number of new classes of glucose-lowering medications, change in weight between baseline and the most recent weight recorded, mean HbA1c after the index date, initiation of insulin, initiation of dialysis, hospitalization, and mortality. To define incident dialysis, we extracted codes used to describe medical services for billing and documentation (ICD-10 and Current Procedural Terminology [CPT] codes) which were selected by clinicians (PLW, EJB) and verified by the research team. The study was approved by the institutional review board at VA Puget Sound Health Care System with the requirement for informed consent waived. Data were accessed on 22 November 24, and study authors did not have access to information that could be used to identify individual participants

## Statistical analyses

We examined distributions of covariates according to exposure status. Depending on the outcome of interest, we fit Poisson, linear, or logistic regression models examining associations of Long COVID documentation with outcomes, and reported incidence rate ratios (IRR), coefficients ($\beta$), and odds ratios (OR) with 95% confidence intervals (CI). We adjusted for age, sex at birth, race/ethnicity, BMI, Charlson comorbidity index, hypertension, dyslipidemia, number of primary care visits in the year prior, urban/rural residence, number of SARS-CoV-2 vaccinations received, and VISN. We assessed for first-order multiplicative interactions of U09.9 with current use of insulin at the index date. We used multiple imputation with 10 sets of imputations for analyses that included BMI due to approximately 20% missing values. To evaluate representativeness and robustness of the imputed values, we examined distributions of imputed and non-imputed BMI and compared values at the 10th, 25th, 50th, 75th, and 90th percentiles. Imputed values did not differ from non-imputed values by more than 0.4 kg/m². In all cases but one, the imputed values did not differ from each other by more than 0.1 kg/m². A

p-value <0.05 was considered significant, and no correction for multiple testing was performed. Analyses were performed using Stata version 18.

## Results

We identified 1,896,080 Veterans with prevalent diabetes who were active in VA care (69.7 years old on average and 6% female). Of these, 1,875,642 (99%) did not have documentation of U09.9 and 20,438 (1%) had ≥ 1 instance of U09.9 documentation (Table 1). Respectively, compared to Veterans without U09.9, those with U09.9 were more likely to report female sex at birth (8% vs. 6%), less likely to report Black race (17% vs. 21%), more likely to report White race (72% vs. 69%), and more likely to report Latinx ethnicity (15% vs. 7%). Veterans with U09.9 had higher mean Charlson comorbidity index (3.5 vs. 2.5), more primary care visits (8% vs. 3% with ≥20 visits in the prior year), and fewer SARS-CoV-2 vaccine doses (69% vs. 77% with ≥2 doses documented). Mean values of age, HbA1c, and BMI were quantitatively similar. The prevalence of patients with no prior positive SARS-CoV-2 test and a positive long COVID diagnosis was low. A total of 75,944 individuals (0.75%) had U09 without a prior positive test for SARS-CoV-2."

Among Veterans without and with Long COVID documentation included in the analysis, median follow-up was 693 days (interquartile range 548–818 days) and 694 days (interquartile range 548–819) respectively. Compared to Veterans without U09.9, those with U09.9 had on average more primary care visits after the index date (14.1 vs 8.1), more new classes of glucose-lowering medications prescribed (0.4 vs. 0.3), and a lesser weight loss (-1.5 kg vs. -4.2.0 kg). Those with U09.9 were more likely to initiate insulin (3.4% vs. 2.6%), initiate dialysis (1% vs. 0%), be hospitalized (28% vs. 13%), or die (13% vs. 12%). Mean HbA1c was similar in the groups (7.1%).

Associations of U09.9 with clinical outcomes are shown in Table 2. Compared to Veterans without U09.9, those with U09.9 had a 36% greater number of primary care visits per year after the index date (IRR 1.36, 95%CI 1.36; 1.37), a 21% greater number of glucose-lowering medication classes added per year (IRR 1.21 95%CI 1.18; 1.24), a 0.62 kg greater weight gain (95%CI 0.52; 0.72), a 0.10-point lower mean HbA1c (95%CI -0.12; -0.08), a 43% greater odds of starting insulin (OR 1.43, 95%CI 1.32; 1.54), a 34% greater odds of starting dialysis (OR 1.34, 95%CI 1.11; 1.62), a 102% greater odds of hospitalization in VA (OR 2.02, 95%CI 1.95; 2.09), and a 13% lower odds of mortality (OR 0.87, 95%CI 0.83; 0.91). Comparing insulin users and non-users, there were some modest differences in the magnitude of associations of U09.9 documentation with outcomes, but in general, the direction of the associations reported was similar in users and non-users. One association did differ substantially in magnitude between current insulin users and non-users: In current insulin users, Long COVID documentation was associated with a 12% greater odds of initiating dialysis (OR 1.12 95%CI 0.85; 1.47), while in non-users, Long COVID documentation was associated with a 67% greater odds of initiating dialysis (OR 1.67, 95%CI 1.29; 2.17) (Table 3).

## Discussion

Among approximately 1.8 million Veterans with diabetes, Long COVID documentation was present in 1% during the time period of this study. Long COVID documentation was associated with significantly greater glucose-lowering medication use, greater odds of starting dialysis, and greater odds of hospitalization, but with lower odds of mortality. Counter to our hypothesis, individuals with Long COVID documentation did not have more weight gain or higher HbA1c, suggesting that adverse effects of Long COVID on kidney function in persons with diabetes may be due to other factors. Among individuals using insulin at baseline, associations of Long COVID documentation with dialysis and mortality were not statistically significant. Future studies should examine mechanisms by which Long COVID might contribute to progression of kidney disease in persons with diabetes.

The current findings are consistent with, and extend, previous research in this area. Large-scale studies have shown associations of SARS-CoV-2 infection with adverse kidney outcomes including chronic kidney disease, with a graded increase in severity according to the severity of the acute infection [11]. In Veterans, compared to non-infected controls,

**Table 1. Characteristics of VHA Veterans with prevalent diabetes, stratified by the presence of Long COVID documentation, October 1, 2021–October 1, 2023.**

| | No Long COVID documentation | | Long COVID documentation | |
|---|---|---|---|---|
| **n** | **1,875,642** | | **20,438** | |
| Age, years | 69.8 | ±11.8 | 68.5 | ±12.0 |
| Age category, years | | | | |
| <40 | 28,067 | 1% | 298 | 1% |
| 40–49 | 84,653 | 5% | 1185 | 6% |
| 50–59 | 233,362 | 12% | 3,082 | 15% |
| 60–69 | 436,869 | 23% | 4,857 | 24% |
| 70–79 | 775,559 | 41% | 8,023 | 39% |
| ≥80 | 317,132 | 17% | 2,993 | 15% |
| Female sex at birth | 112,328 | 6% | 1,655 | 8% |
| Race/ethnicity | | | | |
| Black | 394,539 | 21% | 3,467 | 17% |
| Latinx | 138,516 | 7% | 3,073 | 15% |
| White | 1,285,712 | 69% | 14,757 | 72% |
| Other | 68,800 | 4% | 753 | 4% |
| HbA1c in the year prior to the index date, % | 7.1 | ±1.5 | 7.1 | ±1.5 |
| BMI, kg/m$^2$ | 31.4 | ±6.4 | 32 | ±6.8 |
| BMI category, kg/m$^2$ | | | | |
| <24 | 231,318 | 12% | 2,522 | 12% |
| 24-29 | 516,480 | 28% | 5,598 | 27% |
| 30-34 | 480,747 | 26% | 5,753 | 28% |
| 35-39 | 254,962 | 14% | 3,380 | 17% |
| ≥40 | 151,715 | 8% | 2,255 | 11% |
| Missing | 240,420 | 13% | 930 | 5% |
| Charlson comorbidity index | 2.5 | ±2.4 | 3.5 | ±2.8 |
| Hypertension | 966,279 | 52% | 15,143 | 74% |
| Dyslipidemia | 773,389 | 41% | 13,296 | 65% |
| Primary care visits in the year prior to the index date, # | | | | |
| 0 | 159,717 | 9% | 544 | 3% |
| 1–9 | 1,407,892 | 75% | 12,833 | 63% |
| 10–19 | 252,793 | 13% | 5,324 | 26% |
| ≥20 | 55,240 | 3% | 1,737 | 8% |
| Residence | | | | |
| Not reported | 534,291 | 28% | 6,608 | 32% |
| Urban | 847,196 | 45% | 9,153 | 45% |
| Rural | 494,155 | 26% | 4,677 | 23% |
| SARS-CoV-2 vaccine doses received prior to the index date, # | | | | |
| 0 | 348,354 | 19% | 5,201 | 25% |
| 1 | 83,831 | 4% | 1,114 | 5% |
| ≥2 | 1,443,457 | 77% | 14,123 | 69% |
| **Clinical Outcomes** | | | | |
| Primary care visits after the index date, # | 8.1 | ±8.9 | 14.1 | ±13.3 |
| New glucose-lowering medication classes prescribed, # | 0.3 | ±0.7 | 0.4 | ±0.7 |
| Change in weight, kg | -2.0 | ±6.8 | -1.5 | ±7.4 |

*(Continued)*

**Table 1.** (Continued)

| | No Long COVID documentation | | Long COVID documentation | |
|---|---|---|---|---|
| **n** | **1,875,642** | | **20,438** | |
| Mean HbA1c during follow-up, % | 7.1 | ±1.3 | 7.1 | ±1.3 |
| Initiated insulin, % yes | 48,901 | 2.6% | 703 | 3.4% |
| Initiated dialysis, % yes | 5,536 | 0% | 113 | 1% |
| Hospitalized, % yes | 245,068 | 13% | 5,805 | 28% |
| Died, % yes | 231,356 | 12% | 2,704 | 13% |

Data are presented as mean ± standard deviation, SD for continuous variables and n, column % for categorical variables

survivors of SARS-Cov-2 infection had higher risks of acute kidney injury and end-stage renal disease, as well as declines in estimated glomerular filtration rate (eGFR) [12]. The current findings are generally consistent with this previous body of work. With regard to Long COVID in particular, in an EHR cohort derived from UK National Health Service data (n~319,000), Lin et al. recently reported that Long COVID documentation was associated with significantly greater health care utilization and costs (OR for "use of healthcare resources" 8.29, 95%CI 7.74; 8.87). In 2023, Atiquzzaman et al. reported a 3.4% decline in eGFR among 5,600 patients seen in Long COVID clinics in British Columbia [13], which they note is substantially greater than expected age-related declines. Among those with prevalent diabetes, they found an eGFR decline of 6.2%; however, the proportion who initiated dialysis was not reported. To our knowledge, previous studies have not examined such associations (and/or potential mediators such as changes in weight or hyperglycemia) among persons with diabetes.

At the outset, we hypothesized that Long COVID symptoms might lead to weight gain and/or worsened hyperglycemia via decreases in physical activity and/or impaired cognition; however, we did not find this to be the case. It is thus unlikely that adverse glycemia or metabolic health is the driver of the adverse clinical outcomes we found (higher odds of dialysis and hospitalization) among persons with U09.9 documentation. How Long COVID might contribute to these outcomes is unclear. A number of mechanisms have been hypothesized to contribute to kidney disease after COVID including direct viral infection, endothelial dysfunction, and/or fibrosis [14,15]. This is an area that merits future study. Unexpectedly, we also found that U09.9 was associated with lower odds of mortality. These results are consistent with previous work in VA data by Iwashyna et al., who found no evidence of excess late mortality after SARS-CoV-2 in Veterans [16]. Explanations for this phenomenon are unclear but may be due in part to a depletion of individuals in the cohort who were at high risk of death due to early mortality during or after their initial SARS-CoV-2 infection. Finally, we did not see associations between Long COVID documentation and hyperglycemia or weight. There are several possible reasons for this lack of association. One possibility is that Long COVID truly does not worsen these parameters. Alternatively, true effects of Long COVID on hyperglycemia or weight may have been offset in our data by this group's greater engagement with primary care and escalating diabetes treatment.

Our study has some strengths, most notably a large national sample of racially diverse adults. There are also some important limitations. First, findings may not be generalizable to populations that differ demographically from persons receiving care from the VA or to populations not seeking health care. We relied on documentation of Long COVID in the EHR, which may result in an un underestimate of Long COVID [17] and might explain some of the null associations reported (e.g., weight change, HbA1c). In addition, there were no data on which Long COVID symptoms were present; thus, we could not specifically examine association of impaired exercise tolerance, post-exertional malaise, or Long COVID-related cognitive impairment to the outcomes of interest, nor could we examine the role of physical activity in mediating the associations we observed. We did not distinguish between type 1 and type 2 diabetes, an important

**Table 2. Associations of Long COVID documentation with clinical outcomes among Veterans with diabetes, n = 1,896,080.**

| | Number of primary care visits after the index date | | Number of new glucose-lowering drug classes prescribed | | Change in weight, kg | | Mean HbA1c | | Initiated insulin | | Initiated dialysis | | Hospitalized | | Died | |
|---|---|---|---|---|---|---|---|---|---|---|---|---|---|---|---|---|
| | IRR | 95% CI | IRR | 95% CI | β | 95% CI | β | 95% CI | OR | 95% CI | OR | 95% CI | OR | 95% CI | OR | 95% CI |
| Long COVID documentation, y | 1.36 | 1.36,1.37 | 1.21 | 1.18,1.24 | 0.62 | 0.52,0.72 | -0.10 | -0.12,-0.08 | 1.43 | 1.32,1.54 | 1.34 | 1.11,1.62 | 2.02 | 1.95,2.09 | 0.87 | 0.83,0.91 |
| Age category, years (ref = 50–59 years) | | | | | | | | | | | | | | | | |
| <40 | 0.93 | 0.93,0.94 | 1.09 | 1.08,1.11 | 0.34 | 0.24,0.44 | 0.18 | 0.17,0.20 | 1.70 | 1.61,1.79 | 0.74 | 0.54,1.01 | 1.20 | 1.15,1.25 | 0.45 | 0.40,0.50 |
| 40–49 | 0.96 | 0.96,0.96 | 1.10 | 1.09,1.11 | 0.13 | 0.07,0.19 | 0.10 | 0.09,0.11 | 1.20 | 1.15,1.25 | 0.96 | 0.81,1.13 | 0.96 | 0.94,0.99 | 0.58 | 0.54,0.61 |
| 60–69 | 1.00 | 0.99,1.00 | 0.80 | 0.79,0.80 | -0.52 | -0.56,-0.48 | -0.24 | -0.25,-0.24 | 0.79 | 0.77,0.81 | 1.00 | 0.91,1.09 | 1.16 | 1.15,1.18 | 2.16 | 2.11,2.22 |
| 70–79 | 0.95 | 0.95,0.95 | 0.63 | 0.62,0.63 | -1.05 | -1.08,-1.01 | -0.44 | -0.45,-0.44 | 0.56 | 0.55,0.58 | 0.86 | 0.79,0.94 | 0.98 | 0.97,1.00 | 3.42 | 3.33,3.51 |
| ≥80 | 0.81 | 0.81,0.81 | 0.40 | 0.39,0.40 | -1.78 | -1.82,-1.73 | -0.53 | -0.54,-0.52 | 0.43 | 0.41,0.44 | 0.61 | 0.55,0.69 | 0.91 | 0.89,0.92 | 9.23 | 8.99,9.47 |
| Female sex at birth | 1.13 | 1.13,1.13 | 0.87 | 0.86,0.88 | 0.09 | 0.04,0.14 | -0.24 | -0.25,-0.23 | 0.80 | 0.77,0.83 | 0.70 | 0.61,0.80 | 0.94 | 0.92,0.96 | 0.65 | 0.63,0.67 |
| Race/ethnicity | | | | | | | | | | | | | | | | |
| Black | 1.04 | 1.04,1.04 | 0.81 | 0.80,0.82 | 0.04 | -0.01,0.09 | -0.03 | -0.04,-0.02 | 0.73 | 0.70,0.75 | 1.53 | 1.37,1.71 | 1.53 | 1.50,1.56 | 0.85 | 0.83,0.87 |
| Latinx | 1.02 | 1.01,1.02 | 0.96 | 0.95,0.97 | 0.08 | 0.04,0.12 | 0.08 | 0.07,0.09 | 0.95 | 0.91,0.98 | 1.34 | 1.21,1.49 | 1.12 | 1.10,1.14 | 0.74 | 0.72,0.76 |
| White | 1.03 | 1.02,1.03 | 0.93 | 0.92,0.94 | -0.09 | -0.14,-0.05 | -0.06 | -0.07,-0.05 | 0.83 | 0.81,0.86 | 0.94 | 0.85,1.04 | 1.16 | 1.14,1.18 | 1.08 | 1.06,1.10 |
| Other | 0.99 | 0.99,1.00 | 0.95 | 0.93,0.96 | -0.11 | -0.18,-0.04 | 0.05 | 0.04,0.06 | 0.83 | 0.78,0.87 | 1.22 | 1.04,1.42 | 0.86 | 0.84,0.89 | 0.86 | 0.83,0.88 |
| BMI category, kg/m² (ref = 30–34 kg/m²) | | | | | | | | | | | | | | | | |
| <24 | 0.90 | 0.90,0.90 | 0.63 | 0.63,0.64 | 2.21 | 2.18,2.25 | -0.10 | -0.11,-0.09 | 0.96 | 0.92,0.99 | 1.08 | 0.99,1.18 | 1.43 | 1.41,1.45 | 2.23 | 2.20,2.27 |
| 24–29 | 0.96 | 0.96,0.96 | 0.85 | 0.84,0.85 | 0.92 | 0.89,0.94 | -0.03 | -0.03,-0.02 | 0.98 | 0.95,1.01 | 1.02 | 0.95,1.09 | 1.09 | 1.08,1.11 | 1.21 | 1.20,1.23 |
| 35–39 | 1.04 | 1.04,1.04 | 1.12 | 1.11,1.13 | -1.03 | -1.07,-1.00 | -0.01 | -0.02,-0.00 | 1.06 | 1.03,1.09 | 1.06 | 0.97,1.16 | 0.98 | 0.97,1.00 | 0.97 | 0.96,0.99 |
| ≥40 | 1.11 | 1.10,1.11 | 1.20 | 1.19,1.22 | -2.67 | -2.71,-2.62 | -0.06 | -0.07,-0.06 | 1.08 | 1.04,1.12 | 1.23 | 1.11,1.35 | 1.04 | 1.02,1.06 | 1.08 | 1.06,1.11 |
| Charlson comorbidity index | 1.00 | 1.00,1.00 | 0.98 | 0.98,0.98 | -0.12 | -0.13,-0.12 | 0.04 | 0.04,0.04 | 1.01 | 1.00,1.01 | 1.22 | 1.21,1.23 | 1.18 | 1.18,1.18 | 1.26 | 1.26,1.26 |
| Hypertension | 1.15 | 1.15,1.15 | 1.00 | 1.00,1.01 | 0.17 | 0.15,0.20 | -0.06 | -0.07,-0.06 | 0.87 | 0.85,0.89 | 1.78 | 1.66,1.91 | 1.62 | 1.60,1.64 | 0.94 | 0.93,0.95 |
| Dyslipidemia | 1.06 | 1.06,1.06 | 1.01 | 1.01,1.02 | 0.18 | 0.16,0.21 | 0.08 | 0.07,0.08 | 0.88 | 0.86,0.90 | 0.92 | 0.87,0.98 | 1.14 | 1.13,1.15 | 0.76 | 0.75,0.76 |
| Number of primary care visits in the year prior to the index date (ref = 1–9) | | | | | | | | | | | | | | | | |
| 0 | 0.61 | 0.60,0.61 | 1.24 | 1.23,1.25 | -0.37 | -0.44,-0.31 | 0.27 | 0.26,0.28 | 1.89 | 1.84,1.94 | 1.64 | 1.46,1.85 | 1.18 | 1.16,1.21 | 1.30 | 1.28,1.33 |
| 10–19 | 1.77 | 1.77,1.77 | 0.99 | 0.99,1.00 | -0.18 | -0.21,-0.15 | 0.16 | 0.15,0.17 | 1.00 | 0.98,1.03 | 1.26 | 1.18,1.34 | 1.49 | 1.47,1.51 | 1.46 | 1.45,1.48 |
| ≥20 | 2.87 | 2.86,2.87 | 0.91 | 0.90,0.92 | -0.27 | -0.34,-0.21 | 0.11 | 0.10,0.12 | 0.84 | 0.79,0.89 | 1.37 | 1.23,1.52 | 1.96 | 1.92,2.00 | 2.07 | 2.03,2.12 |
| Residence (ref = urban) | | | | | | | | | | | | | | | | |
| None | 1.20 | 1.20,1.21 | 1.22 | 1.21,1.22 | -0.07 | -0.10,-0.05 | 0.04 | 0.04,0.05 | 1.40 | 1.37,1.43 | 1.51 | 1.43,1.61 | 1.54 | 1.53,1.56 | 0.42 | 0.42,0.43 |
| Rural | 1.02 | 1.02,1.02 | 1.06 | 1.06,1.07 | 0.08 | 0.05,0.11 | 0.02 | 0.02,0.03 | 1.02 | 0.99,1.04 | 0.84 | 0.78,0.91 | 0.72 | 0.71,0.73 | 0.95 | 0.94,0.96 |
| Number of SARS–CoV–2 vaccine doses received prior to the index date (ref = 0) | | | | | | | | | | | | | | | | |
| 1 | 1.08 | 1.08,1.08 | 1.09 | 1.08,1.10 | -0.06 | -0.12,-0.00 | -0.05 | -0.06,-0.04 | 1.02 | 0.98,1.07 | 0.96 | 0.83,1.12 | 1.07 | 1.04,1.09 | 0.94 | 0.92,0.96 |
| ≥2 | 1.15 | 1.15,1.16 | 1.11 | 1.10,1.11 | -0.11 | -0.14,-0.08 | -0.21 | -0.22,-0.21 | 1.00 | 0.98,1.03 | 1.03 | 0.95,1.11 | 1.00 | 0.99,1.01 | 0.51 | 0.50,0.51 |

Abbreviations: BMI (body mass index), CI (confidence interval), IRR (incidence rate ratio), OR (odds ratio)

In addition to the covariates listed, models were additionally adjusted for Veterans Integrated Service Network number (region)

**Table 3. Associations of Long COVID documentation with clinical outcomes among Veterans with diabetes who do and do not use insulin, n = 1,896,080.**

| | Current insulin users n = 521,812 | | Current non-users n = 1,374,268 | | p for interaction* |
|---|---|---|---|---|---|
| | Estimate | 95% CI | Estimate | 95% CI | |
| Number of primary care visits after the index date | 1.37 | 1.37,1.38 | 1.36 | 1.35,1.36 | <0.001 |
| Number of new glucose-lowering drug classes prescribed | 1.26 | 1.21,1.31 | 1.19 | 1.16,1.23 | 0.665 |
| Change in weight, pounds | 0.51 | 0.33,0.69 | 0.65 | 0.53,0.77 | 0.255 |
| Mean HbA1c, % | -0.04 | -0.07,-0.00 | -0.09 | -0.11,-0.07 | <0.001 |
| Odds of initiating dialysis | 1.12 | 0.85,1.47 | 1.67 | 1.29,2.17 | 0.020 |
| Odds of hospitalization | 2.00 | 1.89,2.11 | 2.02 | 1.94,2.11 | 0.280 |
| Odds of mortality | 0.84 | 0.78,0.90 | 0.90 | 0.85,0.95 | <0.001 |

*From model including insulin users and non-users

Models were adjusted for age, sex at birth, race/ethnicity, BMI, Charlson comorbidity index, hypertension, dyslipidemia, number of primary care visits in the year prior, urban/rural residence, number of SARS-CoV-2 vaccinations received, and VISN (region)

limitation. Lastly, this research was conducted in a predominantly male population. Despite this fact, the population included a large number of women (n = 113,983) due to the large overall sample size.

In conclusion, among 1.8 million Veterans with diabetes, Long COVID documentation was associated with greater medication use and higher odds of starting dialysis and hospitalization, but with lower odds of mortality. Individuals with Long COVID documentation did not have more weight gain or higher HbA1c, suggesting that adverse effects of Long COVID on changes in medication use and kidney function in persons with diabetes may be due to other factors Among insulin users, associations of Long COVID documentation with odds of dialysis and mortality were not statistically significant. Future studies should examine mechanisms by which Long COVID might contribute to progression of kidney disease.

## Author contributions

**Conceptualization:** Pandora L. Wander, Edward J. Boyko.

**Formal analysis:** Elliott Lowy.

**Funding acquisition:** Pandora L. Wander.

**Methodology:** Elliott Lowy, Anna Korpak.

**Writing – original draft:** Pandora L. Wander.

**Writing – review & editing:** Elliott Lowy, Anna Korpak, Lauren A. Beste, Edward J. Boyko.

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
