## [Decision Letter · Decision Letter 0]

16 Feb 2025

PONE-D-25-04963Association of U09.9 Long COVID documentation with clinical outcomes among Veterans with diabetesPLOS ONE

Dear Dr. Wander,

Thank you for submitting your manuscript to PLOS ONE. After careful consideration, we feel that it has merit but does not fully meet PLOS ONE’s publication criteria as it currently stands. Therefore, we invite you to submit a revised version of the manuscript that addresses the points raised during the review process.

We look forward to receiving your revised manuscript.

Kind regards,

Gaetano Santulli, MD, PhD

Academic Editor

PLOS ONE

Journal Requirements:

4. Please remove all personal information, ensure that the data shared are in accordance with participant consent, and re-upload a fully anonymized data set. 

Reviewers' comments:

Reviewer's Responses to Questions

**Comments to the Author**

1. Is the manuscript technically sound, and do the data support the conclusions?

Reviewer #1: Yes

Reviewer #2: Yes

Reviewer #3: Partly

2. Has the statistical analysis been performed appropriately and rigorously? 

Reviewer #1: Yes

Reviewer #2: Yes

Reviewer #3: No

3. Have the authors made all data underlying the findings in their manuscript fully available?

Reviewer #1: No

Reviewer #2: Yes

Reviewer #3: No

4. Is the manuscript presented in an intelligible fashion and written in standard English?

Reviewer #1: Yes

Reviewer #2: Yes

Reviewer #3: Yes

5. Review Comments to the Author

Reviewer #1: This is an interesting and well-written paper on a relevant topic. However, a number of issues should be addressed to improve its quality:

General:

- The whole manuscript should be reviewed and if necessary revised according to STROBE guidelines. For example, the study’s design should be indicated with a commonly used term in the title or the abstract.

- International units for height and weight should be used throughout the manuscript and in all tables, e.g. kg instead of pounds.

- Suggest to separate the lower and upper limit of confidence intervals with ";" or "," instead of "-" to make them better readable in case the limits are negative.

- In the spirit of Open and Reproducible Science, the statistical analysis code should be made available in an online repository together with a data dictionary, and the respective URL should be mentioned in the Methods section.

Title:

- Suggest to remove "U09.9" from the title

Abstract:

- "10/1/2021-10/1/2023" is somewhat ambiguous to an international readership and should therefore be revised to e.g. "1 October 2021 - 1 October 2023".

- The data source and the location of the patients included (whole US?) should be mentioned.

- "Veterans were 69.7 years old". Does this refer to 1 Oct 2021?

- The term "index date" needs to be explained.

- All abbreviations such as IRR, CI, OR, VA need to be explained at first appearance, also in the Abstract.

- In the CI of mortality, 8.87 seems to be a typo.

- "These associations do not appear to be mediated through effects on weight or glycemia." This is not supported by the results shown in the Abstract.

Introduction:

- How can the difference in the prevalence of Long-COVID in US veterans (5% vs. 1%) be explained?

- The sentence mentioning reference 6 seems to imply that this reference already contains work from these authors on Long-COVID in insulin users compared to non-users. Is this intended?

- Please explain the term "Veterans". Is this retired military staff? Were all of them engaged in military operations (outside the US), or does this include also e.g. IT personnel?

Methods:

- "Participants were also required to be active in VA care, defined by the presence of ≥1 inpatient or outpatient visit in the prior 12 months." Does this mean that participants were on average less healthy than non-participants (as some of the latter maybe did not need VA care)?

- "For individuals without Long COVID documentation the index date was defined as a random date during a month in which a visit or laboratory test was recorded." This sounds quite arbitrary. Which visit or test was chosen if there was more than one month with a visit or test?

- How was the index date distributed, and what was the median observation period after the index date in individuals with and without Long-COVID? If some differences occur between groups, it might be advisable to add a sensitivity analysis addressing this issue.

- "We collected data..." seems somewhat misleading, as the data had already been collected routinely as I understand it. Should this possibly read "We extracted data..."?

- A reference should be added to the Charlson comorbidity index. A short explanation of this index would be helpful in order to interpret differences in this index between groups.

- What was the rationale to calculate mean HbA1c BEFORE the index date?

- What is a CPT code?

- "11/22/24" should be revised to make as mentioned in a similar comment above.

Results:

- The text would be better readable if veterans without U09.9 were used as reference group and not vice versa (as currently done in the main text).

- The text should be specific in that vaccine doses refer to SARS-CoV-2 vaccinations.

Discussion:

- What is EHR?

- The authors should discuss why they did not differ between type 1 and type 2 diabetes and how this might have affected the results.

- How comes that the participants had a weight loss on average? Would it not be expected to observe an average weight gain over time?

- Is it realistic that only around 2.5% diabetic patients uses insulin? How does this compare to the rate of insulin-dependent diabetes in the US?

- Which associations are referred to by the term "null associations"?

- "These associations do not appear to be mediated through effects on weight or glycemia." Again, this sentence strikes me as this has actually not been shown, which could have been done e.g. by comparing crude and adjusted effect estimates.

Table 1:

- It should be mentioned which tests were used to derive the p-values. However, it appears questionable to report p-values in this table anyway, as also rather irrelevant differences between groups became statistically significant just because of the large sample size.

- It should be mentioned that the % refer to column percent.

- What is the meaning of "Residence: None"? Were approx. 30% of the Veterans homeless?

- Does mean HbA1c refer to before or after the index date? If before, it appears somewhat confusing to present this together with other clinical outcomes which seem to refer to the observation period after th index date.

Table 2:

- It should be mentioned that all covariates were mutually adjusted in each model.

Table 3:

- It should be mentioned that these estimates were adjusted for the same covariates as in Table 2.

Reviewer #2: Review.

Association of U09.9 Long COVID documentation with clinical outcomes among Veterans with diabetes

Abstract;

1. Its inclusive of the objective, relevant methods and key findings.

Introduction

1. The introduction is clearly written and straightforward.

Methods:

1. Are clear and relevant to the study purpose.

Results

1. These are clearly reported in relation to the study objective.

Discussion

1. The discussion starts by summarizing the most important findings. This is good for the reader.

2. The findings are clearly related to other previous related studies.

Remark.

This is generally informative and well written manuscript.

Reviewer #3: The study evaluated the long-COVID clinical outcomes, including mortality, dialysis, glycemia status, and the number of primary care visits in patients with diabetes. The authors found that individuals with long Covid did not have an increased risk of higher HbA1C, weight gain, and mortality, while other clinical outcomes were more likely in them.

The study benefits from a large population of 1.8 million participants. However, the methodology and presentation need significant improvements. Overall, I hope addressing the following comments helps with the manuscript.

1. In the abstract, the OR for mortality in the last line of abstracts result is wrong. Please recheck other variables in the manuscript globally.

2. The term, the impact of long COVID on “glycemia”, used in abstract, highlights, and main text could be changed to better reflect the alterations. I suggest a more precise wording, such as hyperglycemia or glycemia alterations, changes, etc.

3. I suggest adding the study period to the abstract method section.

4. In the abstract, “veterans were 69.7” years needs to be modified.

5. The authors aimed to evaluate the metabolic status of patients with long Covid. However, the full spectrum of metabolic unhealthy was not included, this could significantly limit assessment of the metabolic health.

6. In the method section, it is stated that prior positive Covid test documentation was not necessary, particularly due to the high prevalence of the population with positive rates. Was the prevalence of patients with no prior positive Covid test and a positive long Covid diagnosis checked? How was the potential misclassification assessed?

7. In the method section, I suggest adjusting for hypertension, dyslipidemia, and baseline eGFR as well.

8. In the method section, please determine the timeframe for follow-up. What was the median follow-up period post-index date? How were the variations in follow-up duration between groups accounted for? (For instance, some patients may have had a longer follow-up time for HbA1c changes.)

9. Continuing in my previous question, I suggest providing sensitivity analyses excluding patients with short follow-up periods or insufficient HbA1c data and patients with imputed BMI.

10. In the method section, why was only current insulin use considered as an interaction term? Other medications could potentially modify the effect of Long COVID on metabolic outcomes.

11. Please provide further details on the BMI imputation. How were the robustness and representativeness evaluated?

12. The study is considerably under referenced, and in some instances, the presentation could get better. For the effect of COVID-19 on kidney function, there are multiple large-scale studies with long follow-ups, although not necessarily under the long-covid term. I recommend integrating them into the corresponding section from the discussion.

6. PLOS authors have the option to publish the peer review history of their article (what does this mean? ). If published, this will include your full peer review and any attached files.

**Do you want your identity to be public for this peer review?** For information about this choice, including consent withdrawal, please see our Privacy Policy .

Reviewer #1: No

Reviewer #2: No

Reviewer #3: No

---

## [Author Response · Author response to Decision Letter 0]

28 Mar 2025

28 March 2025

Dear editors:

Thank you for your thoughtful review and for the opportunity to revise this manuscript. We respond to the reviewers’ comments point-by-point below.

Great gratitude,

Luke

Pandora Lucrezia “Luke” Wander, MD, MS, FACP

Associate Professor of Medicine

Adjunct Associate Professor of Epidemiology & Public Health Genetics

University of Washington

Staff Physician

Veterans Affairs Puget Sound Health Care System

Journal Requirements:

We have reviewed the style templates.

We have updated the funding information and financial disclosure sections to match. Importantly, the funding provided was institutional pilot funding. The Seattle VA Office of Research & Development does not provide external numbers for these funds. The funders had no role in study design, data collection and analysis, decision to publish, or preparation of the manuscript.

Data cannot be shared publicly as a condition of approval by the VA Human Research Protection Program at VA Puget Sound. Inquiries may be directed to the VA Puget Sound Research & Development Program at vapugetsoundresearch@va.gov. Data may be available for VA researchers with appropriate approvals.

We have updated the Data Availability statement accordingly

4. Please remove all personal information, ensure that the data shared are in accordance with participant consent, and re-upload a fully anonymized data set.

Please see responses to question 3 above.

Review Comments to the Author

Reviewer #1: This is an interesting and well-written paper on a relevant topic. However, a number of issues should be addressed to improve its quality:

General:

- The whole manuscript should be reviewed and if necessary revised according to STROBE guidelines. For example, the study’s design should be indicated with a commonly used term in the title or the abstract.

We carefully reviewed STROBE guidelines and checklists available at www.strobe-statement.org. We added the following text to the Abstract (p. 2, line 6): “We conducted a retrospective cohort study ….”

- International units for height and weight should be used throughout the manuscript and in all tables, e.g. kg instead of pounds.

We have done so.

- Suggest to separate the lower and upper limit of confidence intervals with ";" or "," instead of "-" to make them better readable in case the limits are negative.

We have done so.

- In the spirit of Open and Reproducible Science, the statistical analysis code should be made available in an online repository together with a data dictionary, and the respective URL should be mentioned in the Methods section.

We very much appreciate this suggestion. Unfortunately, we are not aware of any mechanism within the US Department of Veterans Affairs that permits investigators to upload and share statistical analysis code. Given the federal nature of our institution and the many statutes by which we must abide, we do not have the freedom to share as would be true in a university or research center setting.

Title:

- Suggest to remove "U09.9" from the title

We have done so.

Abstract:

- "10/1/2021-10/1/2023" is somewhat ambiguous to an international readership and should therefore be revised to e.g. "1 October 2021 - 1 October 2023".

We have revised.

- The data source and the location of the patients included (whole US?) should be mentioned.

Yes, this is a national database. We added the following text to the abstract (p. 2, line 6): “We conducted a retrospective cohort study using US Veterans Health Administration electronic health records data to examine …”

- "Veterans were 69.7 years old". Does this refer to 1 Oct 2021?

We added the following text to the Abstract (p. 2, line 14): “Veterans were 69.7 years old on average at cohort entry.”

- The term "index date" needs to be explained.

We added the following text to the Methods (p. 6, line 8): “For individuals with Long COVID documentation the index date was defined as the date of the first instance of U09.9 to appear in the medical record. Individuals without Long COVID documentation were matched by month to those with Long COVID documentation based on the presence of a clinic visit or inpatient discharge that occurred during the same month as the first instance of Long COVID documentation. For individuals without Long COVID documentation, the index date was defined as a randomly selected date during the month in which the clinic visit or discharge occurred.”

- All abbreviations such as IRR, CI, OR, VA need to be explained at first appearance, also in the Abstract.

We have done so.

- In the CI of mortality, 8.87 seems to be a typo.

We have revised this typo. Thanks.

- "These associations do not appear to be mediated through effects on weight or glycemia." This is not supported by the results shown in the Abstract.

We revised the abstract to state (p. 2, line 24): “Individuals with Long COVID documentation did not have more weight gain or higher HbA1c, suggesting that adverse effects of Long COVID on kidney function in persons with diabetes may be due to other factors.”

Introduction:

- How can the difference in the prevalence of Long-COVID in US veterans (5% vs. 1%) be explained?

In some previous studies (including the reference the reviewer is referring to here) were in some cases restricted to individuals with a prior positive test for SARS-CoV-2. Because prior COVID is highly prevalent (>41% of U.S. adults by serology in 2022) and testing patterns are not consistent, in order to increase external validity of findings, in this analysis we did not require documentation of a prior positive COVID test. This difference in study population is the most likely explanation for the difference in prevalences reported.

- The sentence mentioning reference 6 seems to imply that this reference already contains work from these authors on Long-COVID in insulin users compared to non-users. Is this intended?

We appreciate the ambiguity caused by the placement of this reference. We have situated it more appropriately after the following text (p. 5, line 19): “Further, because insulin use necessitates glucose monitoring, and in some cases dose adjustments and/or timing of injections relative to meals and physical activity, the cognitive demands required to manage insulin use may be relatively greater than the demands to manage diet and non-insulin glucose-lowering medications (6).”

- Please explain the term "Veterans". Is this retired military staff? Were all of them engaged in military operations (outside the US), or does this include also e.g. IT personnel?

According to 38 US Code 101, Veterans include all persons who served in the active US military, naval, air, or space force service, and who were discharged or released therefrom under conditions other than dishonorable. Veteran time in service may or may not involve international operations or engagement in combat.

Methods:

- "Participants were also required to be active in VA care, defined by the presence of ≥1 inpatient or outpatient visit in the prior 12 months." Does this mean that participants were on average less healthy than non-participants (as some of the latter maybe did not need VA care)?

Going to the doctor more may mean you are sicker, but it may also mean that you simply have more opportunities to undergo clinical tests and receive diagnoses than people who have fewer doctor visits. We added the following text to the Discussion section (p. 10, line 28): “Findings may not be generalizable to populations that differ demographically from persons receiving care from the VA or to populations not seeking health care.”

- "For individuals without Long COVID documentation the index date was defined as a random date during a month in which a visit or laboratory test was recorded." This sounds quite arbitrary. Which visit or test was chosen if there was more than one month with a visit or test?

If more than one visit or test occurred during the month of interest, the visit or test used was chosen at random from among these. We added the following text to the Methods to clarify how the index date was defined (p. 6, line 8): “For individuals with Long COVID documentation the index date was defined as the date of the first instance of U09.9 to appear in the medical record. Individuals without Long COVID documentation were matched by month to those with Long COVID documentation based on the presence of a clinic visit or inpatient discharge that occurred during the same month as the first instance of Long COVID documentation. For individuals without Long COVID documentation, the index date was defined as a randomly selected date during the month in which the clinic visit or discharge occurred.”

- How was the index date distributed, and what was the median observation period after the index date in individuals with and without Long-COVID? If some differences occur between groups, it might be advisable to add a sensitivity analysis addressing this issue.

This is a good point. The median observation period was similar between the groups. We added the following text to the Results section (page 8, line 14): “Among Veterans without and with Long COVID documentation included in the analysis, median follow-up was 693 days (interquartile range 548–818 days) and 694 days (interquartile range 548–819) respectively.” Given that the median observation periods differed by only one day, we did not feel that this small difference justified a sensitivity analysis.

- "We collected data..." seems somewhat misleading, as the data had already been collected routinely as I understand it. Should this possibly read "We extracted data..."?

Good point. We revised the Methods text to read (p. 6, line 16): “We extracted data …”

- A reference should be added to the Charlson comorbidity index. A short explanation of this index would be helpful in order to interpret differences in this index between groups.

We added the following text to the Methods along with a reference (p. 6, line 19): “We assessed for the presence of chronic comorbidities using the Charlson comorbidity index (which includes history of myocardial infarction, heart failure, peripheral vascular disease, stroke/transient ischemic attack, dementia, chronic obstructive pulmonary disease, connective tissue disease, peptic ulcer disease, liver disease, chronic kidney disease, solid tumor, leukemia, lymphoma, or acquired immune deficiency syndrome).”

- What was the rationale to calculate mean HbA1c BEFORE the index date?

Mean HbA1c calculated before the index date was included to provide a descriptive statistic for glycemic control in the cohort at baseline.

- What is a CPT code?

These are numerical codes used to describe medical services for billing and documentation. We added the following text to the Methods (p. 7, line 7): “To define incident dialysis, we extracted codes used to describe medical services for billing and documentation (ICD-10 and Current Procedural Terminology [CPT] codes) …”

- "11/22/24" should be revised to make as mentioned in a similar comment above.

We have revised the format of this date as requested.

Results:

- The text would be better readable if veterans without U09.9 were used as reference group and not vice versa (as currently done in the main text).

We have revised the text as requested (p. 8, line 4): “Respectively, compared to Veterans without U09.9, those with U09.9 were more likely to report female sex at birth (8% vs. 6%), less likely to report Black race (17% vs. 21%), more likely to report White race (72% vs. 69%), and more likely to report Latinx ethnicity (15% vs. 7%). Veterans with U09.9 had higher mean Charlson comorbidity index (3.5 vs. 2.5), more primary care visits (8% vs. 3% with ≥10 visits in the prior year), and fewer SARS-CoV-2 vaccine doses (69% vs. 77% with ≥2 doses documented). Mean values of age, HbA1c, and BMI were quantitatively similar.

Compared to Veterans without U09.9, those with U09.9 had more primary care visits after the index date (14.1 vs 8.1), more new classes of glucose-lowering medications prescribed (0.4 vs. 0.3), and a lesser weight loss (-1.5 kg vs. -2.0 kg). Those with U09.9 were more likely to initiate insulin (3.4% vs. 2.6%), initiate dialysis (1% vs. 0%), be hospitalized (28% vs. 13%), or die (13% vs. 12%). Mean HbA1c was similar in the groups (7.1%).”

- The text should be specific in that vaccine doses refer to SARS-CoV-2 vaccinations.

We added this requested wording throughout.

Discussion:

- What is EHR?

EHR stands for electronic health record. We included the following text in the Introduction (p. 4, line 20): “We conducted a retrospective observational cohort study using Veterans Affairs (VA) Veterans Health Administration (VHA) electronic health record (EHR) data.”

- The authors should discuss why they did not differ between type 1 and type 2 diabetes and how this might have affected the results.

T1D typically renders individuals ineligible for military service in the U.S. Therefore, all T1D in the Veteran population is presumed to be adult-onset T1D. Adult-onset T1D is associated with lower C-peptide levels at presentation. Thus, it may be challenging to distinguish from T2D because “classic” features of severe insulin deficiency (polyuria, polydipsia, weight loss) may not al

---

## [Decision Letter · Decision Letter 1]

30 Apr 2025

Association of U09.9 Long COVID documentation with clinical outcomes among Veterans with diabetes

PONE-D-25-04963R1

Dear Dr. Wander,

We’re pleased to inform you that your manuscript has been judged scientifically suitable for publication and will be formally accepted for publication once it meets all outstanding technical requirements.

Kind regards,

Gaetano Santulli, MD, PhD

Academic Editor

PLOS ONE

Reviewers' comments:

Reviewer's Responses to Questions

**Comments to the Author**

1. If the authors have adequately addressed your comments raised in a previous round of review and you feel that this manuscript is now acceptable for publication, you may indicate that here to bypass the “Comments to the Author” section, enter your conflict of interest statement in the “Confidential to Editor” section, and submit your "Accept" recommendation.

Reviewer #1: All comments have been addressed

Reviewer #3: All comments have been addressed

2. Is the manuscript technically sound, and do the data support the conclusions?

Reviewer #1: Yes

Reviewer #3: Yes

3. Has the statistical analysis been performed appropriately and rigorously? 

Reviewer #1: Yes

Reviewer #3: Yes

4. Have the authors made all data underlying the findings in their manuscript fully available?

Reviewer #1: No

Reviewer #3: No

5. Is the manuscript presented in an intelligible fashion and written in standard English?

Reviewer #1: Yes

Reviewer #3: Yes

6. Review Comments to the Author

Reviewer #1: The authors have sufficiently addressed all my issues. -----------------------------------------------

Reviewer #3: (No Response)

7. PLOS authors have the option to publish the peer review history of their article (what does this mean? ). If published, this will include your full peer review and any attached files.

**Do you want your identity to be public for this peer review?** For information about this choice, including consent withdrawal, please see our Privacy Policy .

Reviewer #1: No

Reviewer #3: No

---

## [Editor Report · Acceptance letter]

PONE-D-25-04963R1

PLOS ONE

Dear Dr. Wander,

I'm pleased to inform you that your manuscript has been deemed suitable for publication in PLOS ONE. Congratulations! Your manuscript is now being handed over to our production team.

Kind regards,

on behalf of

Professor Gaetano Santulli

Academic Editor

PLOS ONE